

# Maximizing the Use of Pandora Data for Scientific Applications

Prajjwal Rawat[1], James H. Crawford[1], Katherine R. Travis[1], Laura M. Judd[1], Mary Angelique G. Demetillo[1], Lukas C. Valin[2], James J. Szykman[2], Andrew Whitehill[2], Eric Baumann[2], Thomas F. Hanisco[3]

[1]NASA Langley Research Center, Hampton, VA, 23681, United States
[2]Environmental Protection Agency Office of Research and Development, Research Triangle Park, NC, 27709, US
[3]NASA Goddard Space Flight Center, Greenbelt, MD 20771, USA

*Correspondence to*: Prajjwal Rawat (prajjwal.rawat@nasa.gov) and James H. Crawford
(james.h.crawford@nasa.gov)

**Abstract.**

As part of the Pandonia Global Network (PGN), Pandora spectrometers are widely deployed

around the world. These ground-based, remote-sensing instruments are federated such that they employ a common algorithm and data protocol for reporting on trace gas column densities and lower atmospheric profiles using two modes based on direct-sun and sky-scan observations. To aid users in the analysis of Pandora observations, the PGN standard quality assurance procedure assigns flags to the data indicating high, medium, and low quality. This work assesses the

suitability of these data quality flags for filtering data in the scientific analysis of nitrogen dioxide ($NO_2$) and formaldehyde (HCHO), two critical precursors controlling tropospheric ozone production. Pandora data flagged as high quality assures scientifically valid data and is often more abundant for direct-sun $NO_2$ columns. For direct-sun HCHO and sky-scan observations of both molecules, large amounts of data flagged as low quality also appear to be valid. Upon closer

inspection of the data, independent uncertainty is shown to be a better indicator of data quality than the standard quality flags. After applying an independent uncertainty filter, Pandora data flagged as medium or low quality in both modes can be demonstrated to be scientifically useful. Demonstrating the utility of this filtering method is enabled by correlating contemporaneous but independent direct-sun and sky-scan observations. When evaluated across 15 Pandora sites in

North America, this new filtering method increased the availability of scientifically useful data by as much as 90% above that tagged as high quality. A method is also developed for combining the direct-sun and sky-scan observations into a single dataset by accounting for biases between the two observing modes and differences in measurement integration times. This combined data provides a more continuous record useful for interpreting Pandora observations against other

independent variables such as hourly observations of surface ozone. When Pandora HCHO



columns are correlated with surface ozone measurements, data filtered by independent uncertainty exhibits similarly strong and more robust relationships than high-quality data alone. These results suggest that Pandora data users should carefully assess data across all quality flags and consider their potential for useful application to scientific analysis. The present study provides a method for maximizing use of Pandora data with expectation of more robust satellite validation and comparisons with ground-based observations in support of air quality studies.

## 1 Introduction

The Pandora spectrometer is a ground-based UV-visible (UV-VIS) remote-sensing instrument utilized to validate space-based UV-VIS sensors and understand local air quality (Herman et al., 2009; Wang et al., 2010; Spinei et al., 2018; Tzortziou et al., 2012; Herman et al., 2019; Szykman et al., 2019; Judd et al., 2019; Verhoelst et al., 2021; Di Bernardino et al., 2023). The Pandora instrument was developed to observe column nitrogen dioxide ($NO_2$) in direct-sun mode, in addition to the more traditional sky-scan observations obtained using multi axial differential optical absorption spectroscopy (MAX-DOAS). Once calibrated, direct-sun observations are less impacted by atmospheric parameters that influence the air mass factor (AMF) and which also cause uncertainty in space-based retrievals, such as the vertical gas profile or surface albedo (Cede et al., 2006; Herman et al., 2009). More recently, Pandora instrument retrievals of column formaldehyde (HCHO) have become available, expanding opportunities to evaluate the importance of both nitrogen oxides ($NO_x$) and volatile organic compounds (VOCs) as precursors of ozone and secondary aerosol. This was enabled by the removal of instrument components made of Delrin which were discovered to outgas and create interferences in the detection of HCHO (Spinei et al., 2021).

The Pandonia Global Network (PGN) is a NASA and ESA-sponsored ground-based network of Pandora spectrometers supporting in situ and remotely sensed air quality monitoring and satellite validation activities (https://blickm.pandonia-global-network.org). The major advantages of the PGN are uniform instrument design, homogeneous calibrations, centralized data monitoring and processing, and real-time data distribution (Herman et al., 2009, 2018; Spinei et al., 2018; Herman et al., 2019; Cede et al., 2021). Pandora instruments are calibrated by NASA Goddard Space Flight Center and deployed to local operators around the world who join the PGN. Consistent retrievals





across the network are performed by Luftblick Earth Observation Technologies and are hosted on the PGN website (Cede et al., 2021). Presently, there are more than 130 Pandora instruments worldwide and over 60 across the U.S. (Szykman et al., 2019; Chang et al., 2022).

Pandora instruments have provided valuable information on $NO_2$ retrieval biases from the polar-orbiting Sentinel-5P TROPOspheric Monitoring Instrument (TROPOMI) (Judd et al., 2020;
Verhoelst et al., 2021; Park et al., 2022; Ialongo et al., 2020) and Ozone Monitoring Instrument (OMI) (Tzortziou et al., 2012; Reed et al., 2015; Herman et al., 2019). The high temporal availability of Pandora data has allowed for deep investigation of local meteorology on column $NO_2$ and how well that is captured by air quality models (Goldberg et al., 2017; Choi et al., 2020; Tzortziou et al., 2022; Wang et al., 2023; Adams et al., 2023; Tzortziou et al., 2023). The relatively
new formaldehyde retrievals from Pandora (Spinei et al., 2018, 2021) are beginning to be incorporated into model and satellite validation efforts (Herman et al., 2018) and show promise for characterizing the local ozone photochemical environment (Schroeder et al., 2016; Travis et al., 2022).

Geostationary satellite observations provide a new challenge in interpreting $NO_2$ and HCHO
columns at unprecedented spatial and temporal scales (Kim et al., 2017; Judd et al., 2018). Pandora instruments will be the main source of validation data for both $NO_2$ and HCHO columns for the Tropospheric Emissions: Monitoring of Pollution (TEMPO) geostationary air quality satellite launched over the U.S. in 2023 (Szykman and Liu, 2023; Zoogman et al., 2017). Pandora instruments are also being used to validate the Geostationary Environmental Monitoring
Spectrometer (GEMS) satellite retrievals over East Asia (Kim et al., 2023). Given the growing need for Pandora data to support satellite observations and model development, improved methods for ensuring data availability and quality are needed. In this work, a data filtering method is presented for assessing suitability of data beyond the standard quality flags provided by PGN. This method takes advantage of the independent but contemporaneous measurement modes of Pandora
operation (direct-sun and sky-scan) to analyze Pandora data quality and maximize the availability of data for scientific application.



## 2 Data and methodology

### 2.1 Pandora measurements and data retrieval

The Pandora spectrometer is an instrument for ground-based remote sensing of a range of trace gases, including $NO_2$, HCHO, $O_3$, and $SO_2$, with temporal resolution on the order of seconds to minutes. Pandora is a passive UV-VIS spectrometer system equipped with a responsive sun tracker that points the optical head sensor to the sun or sky with a spatial accuracy of up to ±0.1°. A basic description of the Pandora instrument is summarized here with additional details provided in supplementary section S1.1 and available in Cede et al., (2021). Briefly, recorded light is transmitted through fiber optical cables to a UV-VIS low stray light spectrometer (Herman et al., 2009; Tzortziou et al., 2012). The spectrometer covers a spectral range of 280-530 nm with a spectral resolution of 0.6 nm. To minimize the dark current noise, the spectrometer is maintained at a stable temperature of 20°C with a thermoelectric cooler. The Pandora spectrometer operates in two daytime viewing geometries, namely, direct-sun (DS) and scattered sunlight or sky-scan (SS) (Figure 1), with varying degrees of temporal resolution spanning from milliseconds to minutes depending on the mode of operation.

The Pandora retrieval involves several sequential steps or data processing levels that are all archived by PGN. Raw measurement spectra are archived as L0. L1 data include corrections for instrumental characteristics such as dark signal, non-linearity, latency, flat field, temperature sensitivity, stray light, and wavelength corrections. L2Fit data includes spectral fitting, which leads to slant column amount calculations relative to the reference spectrum employing DOAS analysis (Herman et al., 2009; Cede, 2022; Gebetsberger et al., 2023a). L2 data results from the conversion from slant columns to vertical columns, lower tropospheric column amounts, surface concentrations, and profiles utilizing geometrical AMFs for direct sun and analytical methods for sky scan, briefly described in the supplement to this paper (section S1.1). The reported accuracy of the DS total vertical column of $NO_2$ is $2.7\times10^{15}$ molecules cm$^{-2}$ (0.1 Dobson unit, where 1 DU = $2.7\times10^{16}$ molecules cm$^{-2}$) with a precision of $2.7\times10^{14}$ molecules cm$^{-2}$ (0.01 DU) (Herman et al., 2009). For HCHO DS a small statistical error up to 6% and notable systematic error up to 26% are reported (Spinei et al., 2018, 2021) while the recent Pandora Delrin-free instruments require more assessment for quantifying accuracy and precision. The Pandora SS measurements still lack a robust validation; however, the CINDI-2 intercomparison included a Pandora (named as NASA





instrument) and reported a bias for measurements in the SS mode of about $-0.05 \times 10^{16}$ molecules
cm$^{-2}$ for HCHO column and about $-0.02 \times 10^{16}$ molecules cm$^{-2}$ for NO$_2$ column against the median
of the participating instruments (Tirpitz et al., 2021; Verhoelst et al., 2021). A brief review on
current Pandora software and hardware changes, retrieval scheme, and data quality flagging are
discussed in supplementary section S1.1.

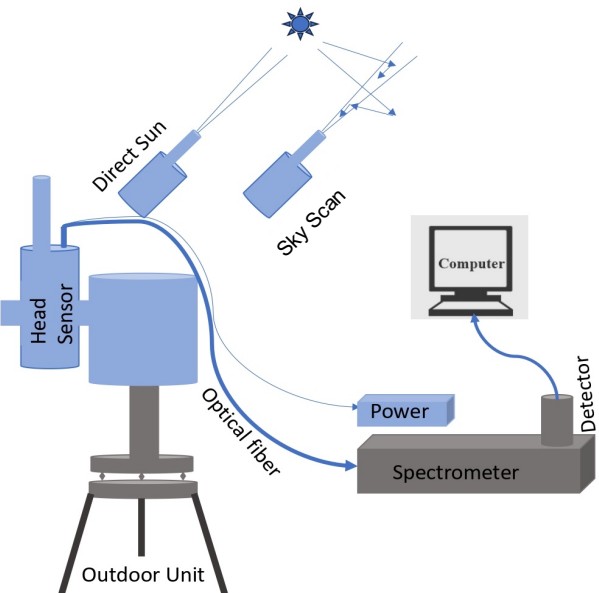

**Figure 1.** Block diagram of a Pandora instrument and its two viewing geometries.

## 2.2 Pandora observational schedule

Pandora instruments are locally operated and mainly run on the standard schedule alternating
between DS and SS mode, although there is an option for user-defined schedules. The DS and SS
modes also alternate between an Open (unfiltered) mode for NO$_2$ and a U340 (filtered for
maximum transmission near 340nm) mode for HCHO observations (Cede et al., 2021; Spinei et
al., 2018). A typical example of the Pandora standard schedule on a clear sky day is shown in
Figure 2a, with the typical total hours per day for each routine annotated. Typically, Pandora
spends about 1.5 hrs each day in DS NO$_2$ (SQ) and DS HCHO (SS, not to be confused with "sky-





scan") modes and about 1.5 hrs each in SS mode for lower tropospheric columns of $NO_2$ and HCHO (EO) and HCHO only (EU). Longer more detailed sky scanning (EL and EK) occurs for a total of about 1-2 hrs for profile observations. Find Sun (FS) routines occur throughout the day, but only take a short time (few seconds) over each hour. Typically, Pandora spends 65% of the

time in SS mode and 30% in DS mode under clear sky conditions. Further details of all Pandora routine acronyms are described in supplementary section S1.2 and the Pandora user manual (Cede et al., 2021). The total hours per day for each routine is annotated in Figure 2a. The inset panels in Figure 2 (b and c) provide a closer view of the schedule for an 18-minute window as the instrument switches between DS and SS modes. The varying effective measurement times ($t_{eff}$) illustrate the

observation frequency for each mode and product. Generally, the Pandora instrument possesses the capability to capture solar spectra at adjustable integration times, ranging from 2.5 ms to 4 s, with an overall measurement duration of approximately 40 s, encompassing a few dark current measurements (Cede et al., 2021). The integration time for DS $NO_2$ (with no filter) under bright sun is about 4 ms, and roughly 4000 spectra are acquired repeatedly and averaged to obtain very

high signal-to-noise ratios with high precision (Herman et al., 2018). For DS HCHO (with U340 filter), a longer integration time (30-1000 ms) is used; hence, there are fewer measurements for HCHO compared to $NO_2$. Typically, in the DS mode, there are about 5 times more observations of $NO_2$ compared to HCHO column per scan (Figure 2c). In the SS mode, zenith measurement scans are routinely collected at a specified azimuth angle, with the north or 0° being the preferred

direction. The zenith angles scans can vary by site with the lowest scan typically occurring between 89° and 85° depending on viewing conditions available at the site. The extended SS measurements are taken at multiple zenith angles (section S1.2) to derive lower tropospheric columns and profiles typically up to 2-4 km.


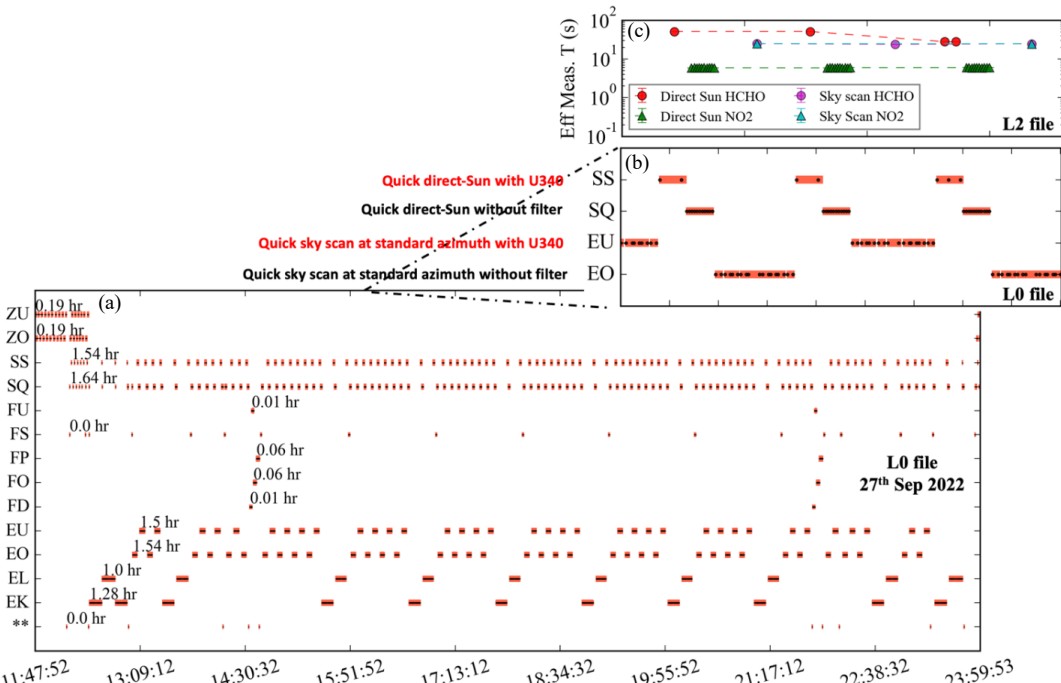

**Figure 2.** (a) Pandora standard schedule of operations (UTC hour) on a clear sky day on 27 Sep 2022 at the University of Houston site for different observing modes with a zoomed-in, 18-minute window for both L0 (b) and L2 (c) file information. The black points in the L0 schedule show the start of each measurement time and underlying red line is the total time in that respective mode. The L2 data (c) shows the alternating retrieval of $NO_2$ (without filter) and HCHO (with U340 filter) column from direct-sun and sky-scan modes.

## 3.0 Pandora data quality flags and an alternative data filtering method

The PGN team developed a software suite to perform retrievals for $NO_2$ and HCHO and provide a measure of data quality to guide users (Cede et al., 2021, 2023). Reduced data quality can arise from instrumental and/or atmospheric sources. Data flags provide information on both the data quality and data assurance level (Gebetsberger et al., 2023b). Data quality is indicated in the unit position, with a 0, 1, or 2 indicating high, medium, and low quality respectively. While data that is preliminary, but not yet quality assured, is useable for science, this status is indicated by a 1 in the tens position. Data that is unusable for whatever reason are indicated by a 2 in the tens position. Thus, data flagged as 0 or 10 are considered useable and high quality, and users are cautioned to pay additional scrutiny to data flagged as 1 or 11 (medium quality). By contrast, users are



discouraged from using any data flagged as 2 or 12 (low quality). These quality flags are based on threshold values of various parameters, i.e., uncertainties, wrms (normalized rms of the fitting residuals weighted with the independent uncertainty), atmospheric variability (a measure of radiance variance in field of view or measure of cloudiness), wavelength shift, and AMF (Table S1). More description of the data quality flags is provided in supplementary section S1.1.

Following the suggestion of the Pandora user manual (Cede et al., 2021, 2023), many previous studies with Pandora v1.8 data utilize only high-quality flag data (0, 10) (e.g., Park et al., 2022; Verhoelst et al., 2021; Liu et al., 2023). However, when using only high-quality flagged data, a large fraction of observations is often eliminated. Figure 3 shows quality flag statistics across the 15 Pandora sites evaluated in this study (listed in Table S2). They have been selected since they

fall under recent NASA airborne campaigns that covered areas of the eastern U.S and Houston, TX. These are the Tracking Aerosol Convection Interactions Experiment-Air Quality (TRACER-AQ) in 2021, the Student Airborne Research Program (SARP-East) in 2023, and the Synergistic TEMPO Air Quality Science (STAQS) study in 2023. Across these sites, high-quality data flags dominate only for DS $NO_2$ observations during the 2021-2022 period, typically being between 60-

80%. For other observations, low quality data flags often dominate, on average accounting for 57%, 45%, and 46% of data for SS $NO_2$, DS HCHO, and SS HCHO, respectively, across the evaluated sites, making much of the data unavailable for supporting field campaign analysis.

The schedule of Pandora operations shown in Figure 2 demonstrates the contemporaneous nature of DS and SS observations. Figure 3 further shows that the quality flags for the two observing

methods can often differ substantially. This enables an independent assessment of data quality by looking at the correlation of contemporaneous (within 5 min) DS and SS observations for different quality flag combinations. An example is provided in Figure 4 showing HCHO observations for the University of Houston Pandora site. Note that there are nine possible combinations of high, medium, and low-quality data pairs. More than half of those data pairs (57%) include one low-

quality observation. Low-quality SS observations correlate as well or better with medium and high-quality data as do data pairings including only medium and high-quality data. There is more scatter in the data that include low-quality DS observations (right column of panels in Figure 4), but there is still evidence of a large number of well correlated data, even for pairings of low-quality



data for both DS and SS modes. Given the independent nature of the DS and SS observations, it is

worthwhile to examine the role of measurement uncertainty in these correlations.

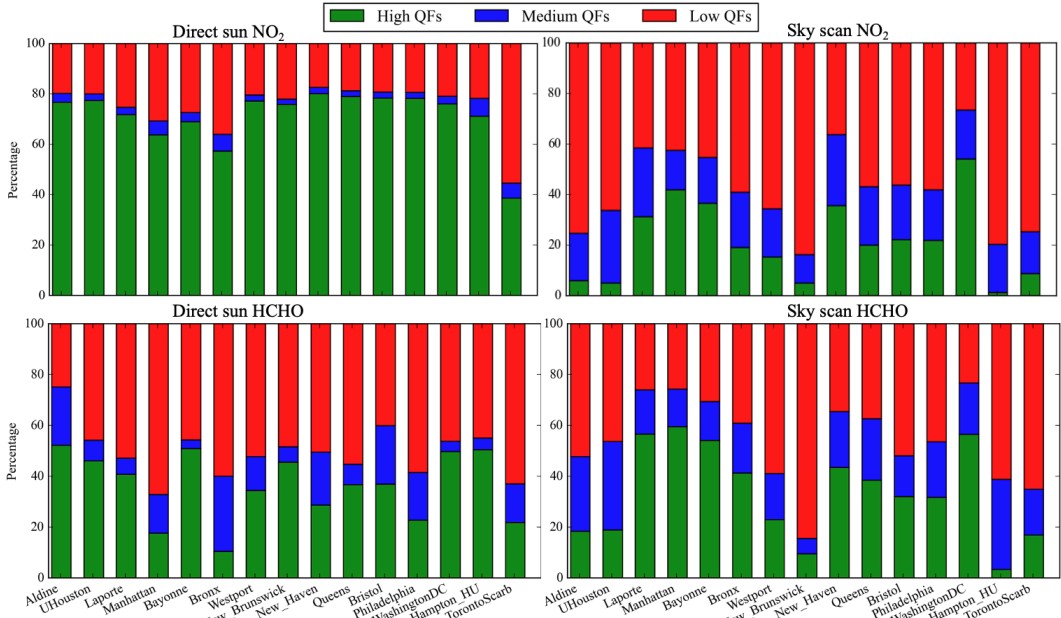

**Figure 3.** Percentage contribution of high (0, 10), medium (1, 11), and low (2, 12) quality flags
for 15 Pandora sites for observations in both DS (left panels) and SS (right panels) modes during
2021-2022. Percentage contributions are shown for $NO_2$ in the top panels and HCHO in the bottom
panels.

Figure 5 shows the frequency distribution of the retrieved independent uncertainty for high,

medium, and low-quality Pandora HCHO and $NO_2$ column observations for both DS and SS mode

for non-negative vertical columns at the University of Houston Pandora site. Independent

uncertainty is a measure of the photon noise across the detector pixels which is propagated to slant

columns for each retrieval and provided in the L2 product (Table S1). Figure 5 shows that there is

significant overlap in the independent uncertainty distribution across all three quality flags for each

measurement mode. The figure inset shows the corresponding cumulative probability distributions

and a cut-off value (black dotted line) that preserves the high-quality data while including a

significant amount of overlapping data flagged as medium or low quality. In the present analysis,

the cut-off value for independent uncertainty is defined as $\mu + 3\sigma$ of the independent uncertainty

distribution for high-quality data, where $\mu$ is mean and $\sigma$ is standard deviation. The gray shaded





area in Figure 5 shows the portion of the data falling outside this limit. For the University of Houston Pandora, the cut-off values are $0.37\times10^{15}$ molecules cm$^{-2}$ for DS independent uncertainty and $1.79\times10^{15}$ molecules cm$^{-2}$ for SS independent uncertainty in HCHO column measurements. For NO$_2$ column measurements the cut-off values are $0.62\times10^{14}$ molecules cm$^{-2}$ for DS independent uncertainty and $1.81\times10^{14}$ molecules cm$^{-2}$ for SS independent uncertainty. In this method, Pandora data having independent uncertainty less than the cut-off limit are considered scientifically useful regardless of their data flag.

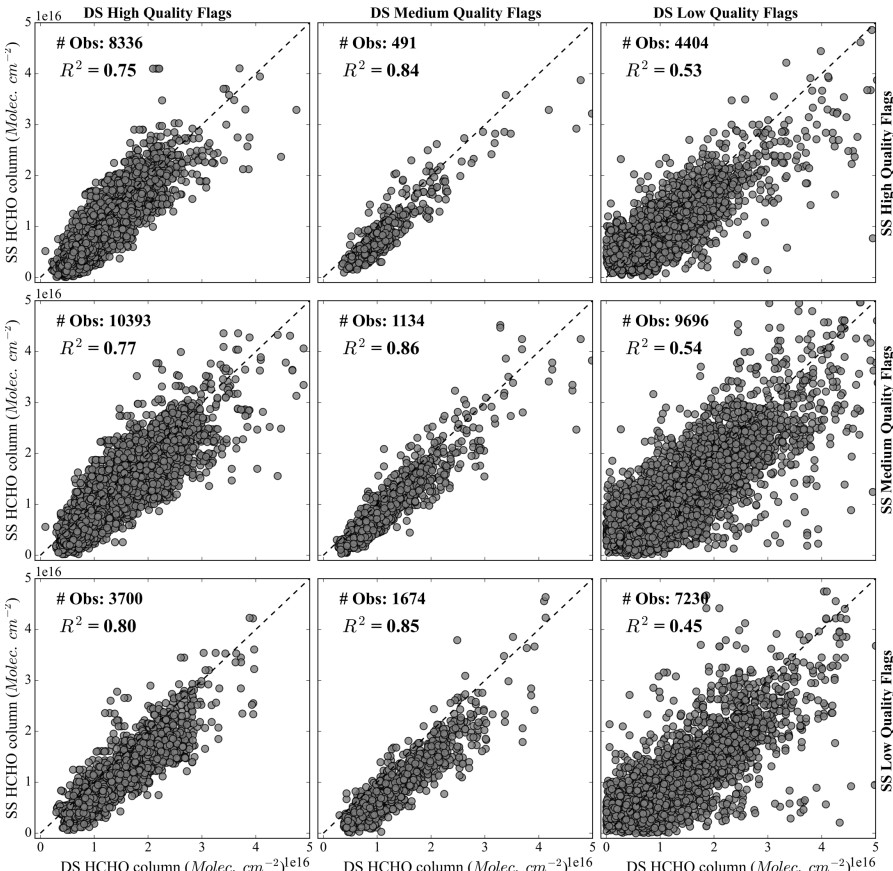

**Figure 4**. Correlation of contemporaneous DS and SS observations of HCHO column for different quality flag combinations. Data is from the University of Houston Pandora site over two years (2021-2022).



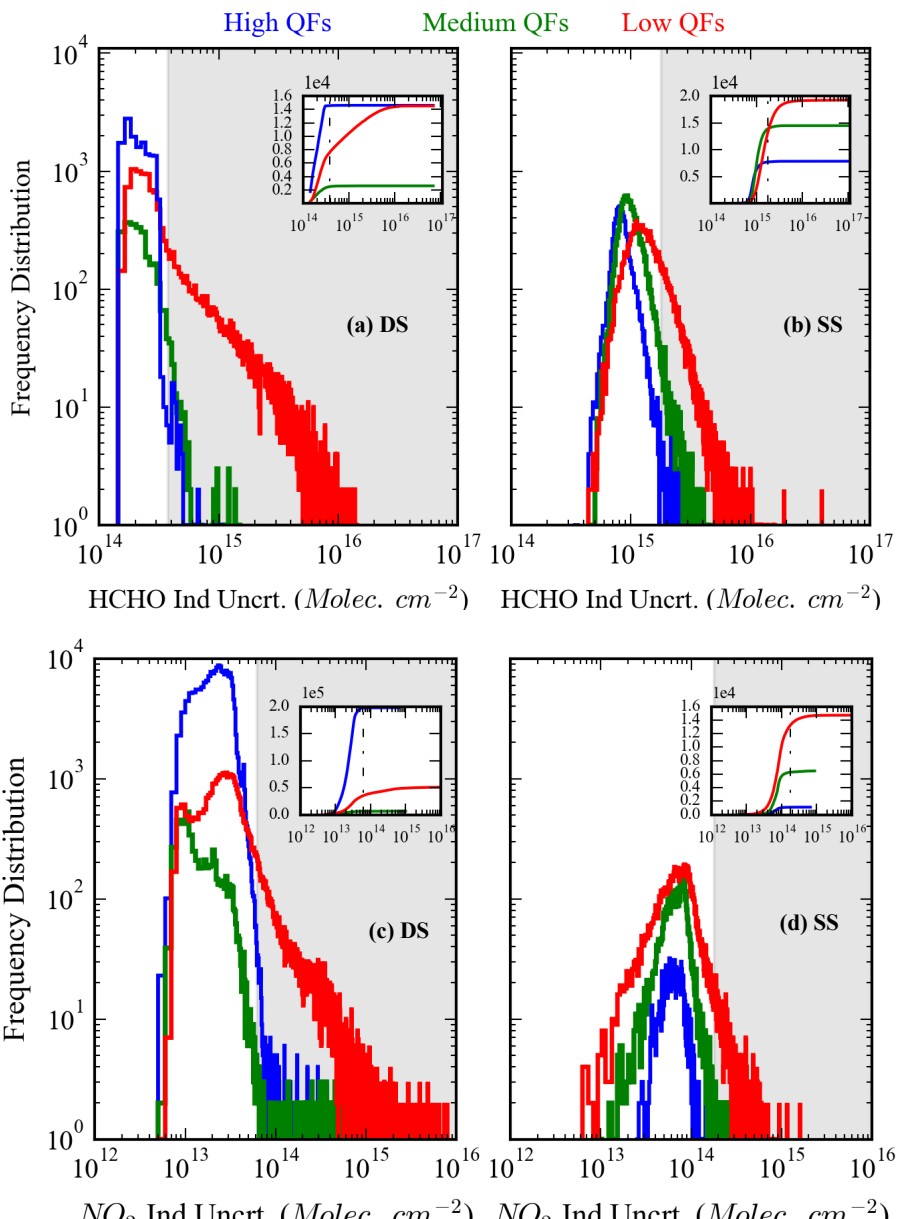

**Figure 5.** Frequency distribution of independent uncertainty in three quality flags: high (blue), medium (green) and low (red) for a) column HCHO direct-sun, b) column HCHO sky-scan, c) column $NO_2$ direct-sun, and d) column $NO_2$ sky-scan observations at the University of Houston Pandora site. The cumulative distributions for each panel are inset with the cut-off values used in the present analysis shown as a black dashed vertical line. Data values exceeding the cut-off are shaded in gray for the frequency distributions.





This proposed data filtering method is evaluated in Figure 6 for HCHO observations at the University of Houston Pandora site, showing how specific data pairs are related to the independent uncertainty cut-off values. Data pairs with uncertainty falling above the cut-off for DS only (blue points) and both DS and SS (red) are almost exclusively limited to the low-quality flagged data

(right column and bottom row of panels in Figure 6). These points also account for most of the scatter in the correlations. Data pairs with uncertainty falling above the cut-off for SS data only (yellow) tend toward larger column amounts. This calls for a second filtering step in which observations with percent independent uncertainties of less than 10% of the column amount are retained even if the absolute independent uncertainty exceeds the cut-off value. This preserves a

small but important set of large column abundances that are important for exploring the full dynamic range of column variability.

Finally, there can be a few rare outliers associated with unusually large wrms values even after applying this independent uncertainty cut-off. For this analysis, the additional handful of points

with wrms > 0.01 are removed. Another variable specific to the SS observations is the maximum horizontal distance (MHzD) which can occasionally reach large values. For this analysis, data with MHzD > 20 km are removed as well. Specific column numbers from Pandora files for the variables used in this analysis are provided in Table S1 to facilitate application of this method.

Figure 7 shows the filtered data for the University of Houston Pandora site colored by data density. After the filtering method is applied, correlations notably improve for combinations that include DS low quality flagged data. Among all pairings ($R^2$ = 0.72 to 0.86), data quality flags no longer appear to be a useful predictor of correlation between contemporaneous SS and DS observations. This suggests that all remaining data may be scientifically useful regardless of quality flag. Figure

7 also shows that there is a bias between SS and DS observations for HCHO column. This is not unexpected given that SS observations are limited to the lower troposphere, but as discussed later in section 3.3, this bias must be addressed for applications where it is advantageous to combine SS and DS observations into a larger unified data set.



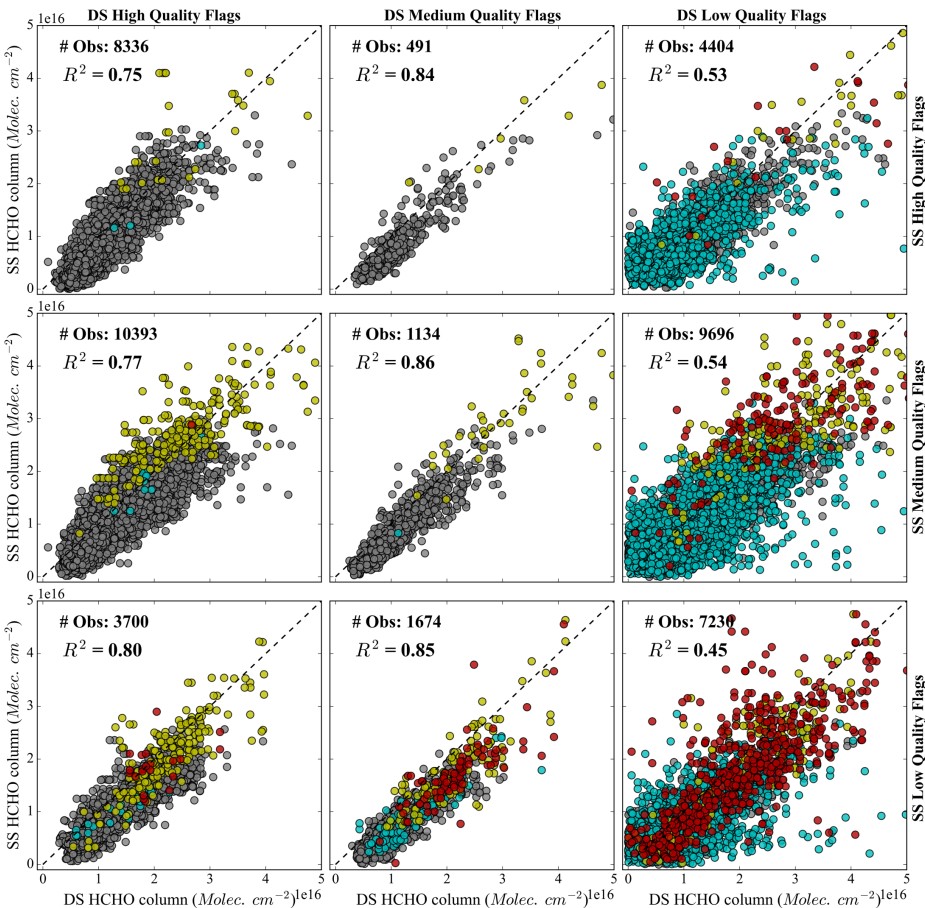


**Figure 6.** Correlation of contemporaneous DS and SS observations of HCHO column for different quality flag combinations at the University of Houston Pandora site over two years (2021-2022). Data points are colored by cutoff values used for HCHO column independent uncertainty values. Gray indicates points for which both DS and SS uncertainties are less than the cut-off, other colors

indicate points for which only DS (cyan), only the SS (yellow), or both DS and SS (red) uncertainties exceed the cut-off value.

Figure 8 shows the same plot as Figure 6 but for $NO_2$ columns at the University of Houston Pandora site. There are more total pairs in this comparison since DS $NO_2$ observations are more

frequent than HCHO (see Figure 2). Again, reasonable correlation is observed in all the quality flags ($R^2$ = 0.68 to 0.73). For $NO_2$, the filtering method leads to very little data being removed, thus correlations are essentially unchanged in Figure 9 for the filtered data. Again, there is a bias





with DS values being greater than SS values. This is mainly due to stratospheric $NO_2$ which is only detected in the DS mode (see discussion in section 3.3).


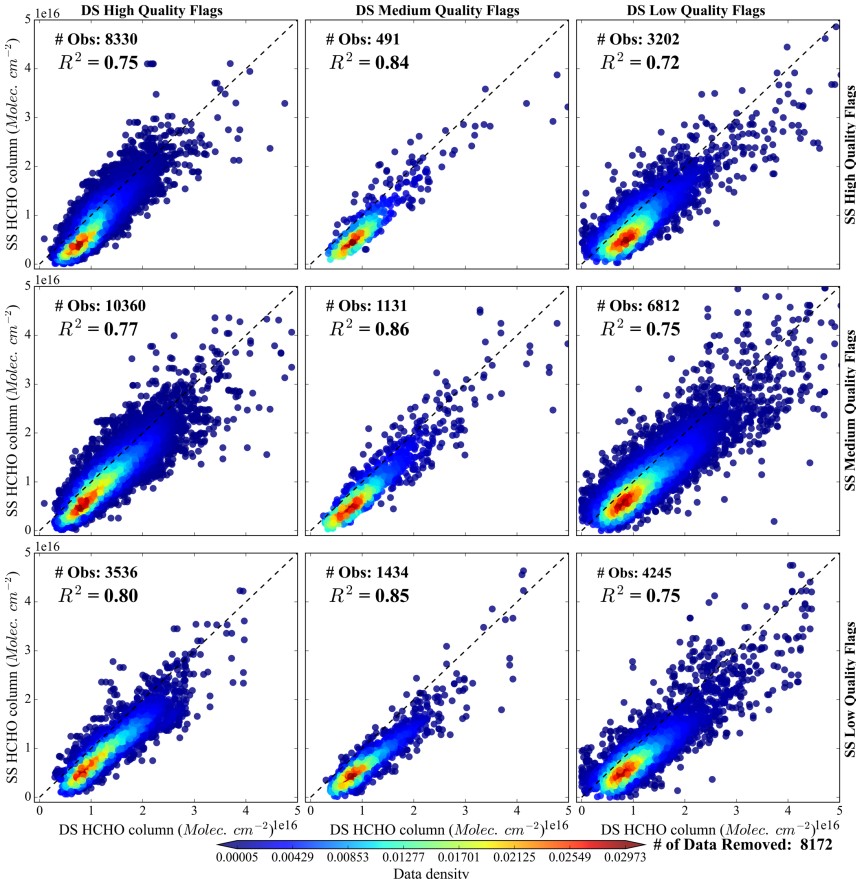

**Figure 7.** Correlation of contemporaneous DS and SS observations of HCHO column for different quality flag combinations after applying new filtering method to the University of Houston Pandora site for 2021-2022. Data points are colored by normalized gaussian kernel density of data.




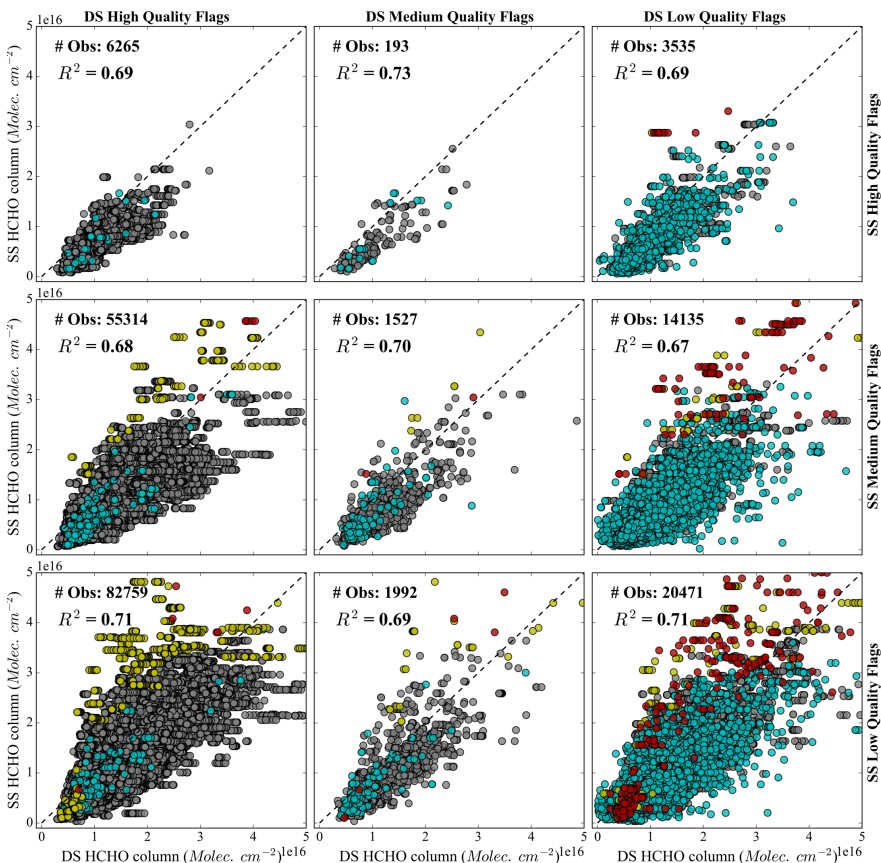

**Figure 8.** Correlation of contemporaneous DS and SS observations of $NO_2$ column for different quality flag combinations at the University of Houston Pandora site for 2021 – 2022. Data points are colored by cutoff values used for $NO_2$ column independent uncertainty values. Gray indicates points for which both DS and SS uncertainties are less than the cut-off, other colors indicate points for which only DS (cyan), only the SS (yellow), or both DS and SS (red) uncertainties exceed the cut-off value.





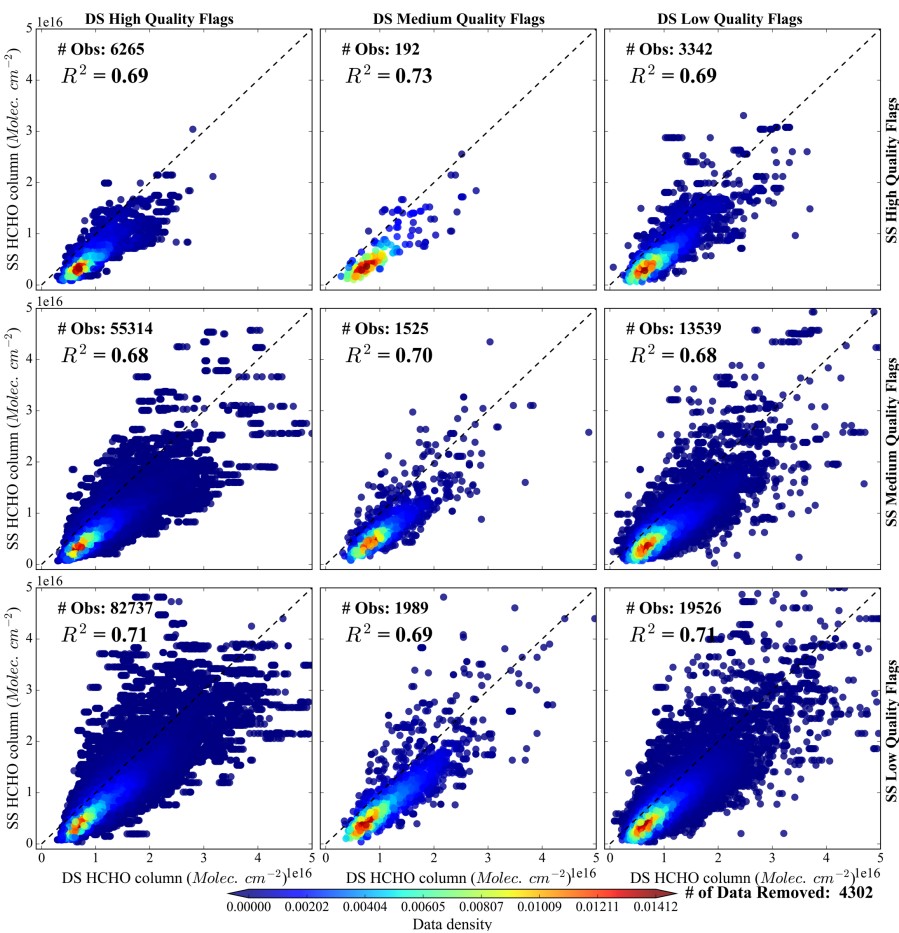

**Figure 9.** Correlation of contemporaneous DS and SS observations of NO₂ column for different quality flag combinations after applying new filtering method to the University of Houston Pandora site for 2021-2022. Data points are colored by normalized gaussian kernel density of data.





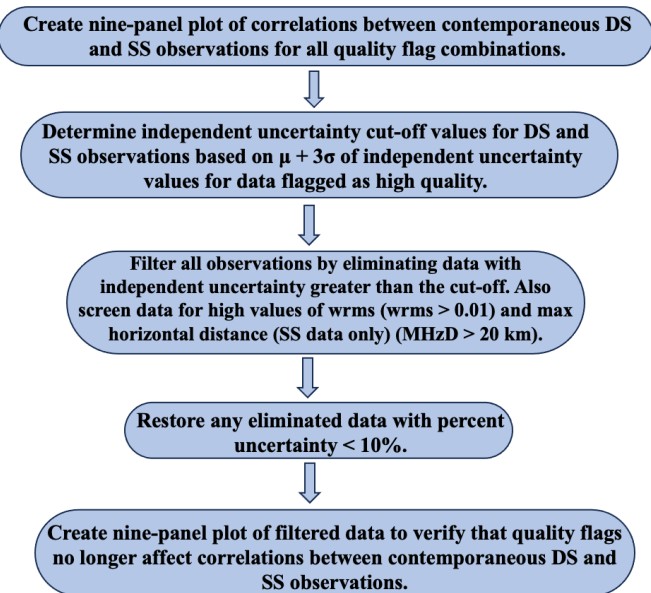

**Figure 10.** New data filtering method used in the present analysis to increase the scientific availability of Pandora observations.

The method demonstrated in Figures 4 through 9 is summarized in Figure 10. It can be applied to any PGN site with contemporaneous DS and SS observations. Results of applying the method are show in Figure 11 for the 15 Pandora sites falling under recent NASA airborne campaigns. Data from 2021 and 2022 are used for all sites, although La Porte, Texas only has data after August 2021 to February 2022. The bars in Figure 11 show the occurrences of the high (green), medium (blue), and low (red) quality flagged data across these 15 Pandora sites. The adjacent grey bar shows the amount of usable data after applying the new filtering method. Compared to using only high-quality flagged data, the new filtering method increases available data up to 90% for the HCHO column in both SS and DS modes (Figures 11c and 11d) and for $NO_2$ in SS mode (Figure 11b). Most DS $NO_2$ columns were already flagged as high quality, but the new filtering method still increases available data by 20% to 60% (Figure 11a). The amount of available data varies considerably between sites. This can be due to a number of reasons including instrument down time, use of non-standard observing schedules, and periods of data flagged as unusable (20, 21, or 22).



As noted in Figure 10, the first step in verifying the success of the method is to demonstrate the quality of correlation across all quality flag combinations. To further strengthen the case for using this method, independent observations are used in the following sections to demonstrate more robust scientific analyses. Evaluation of biases between DS and SS observations and prospects for

combining them into a single data set are also explored.

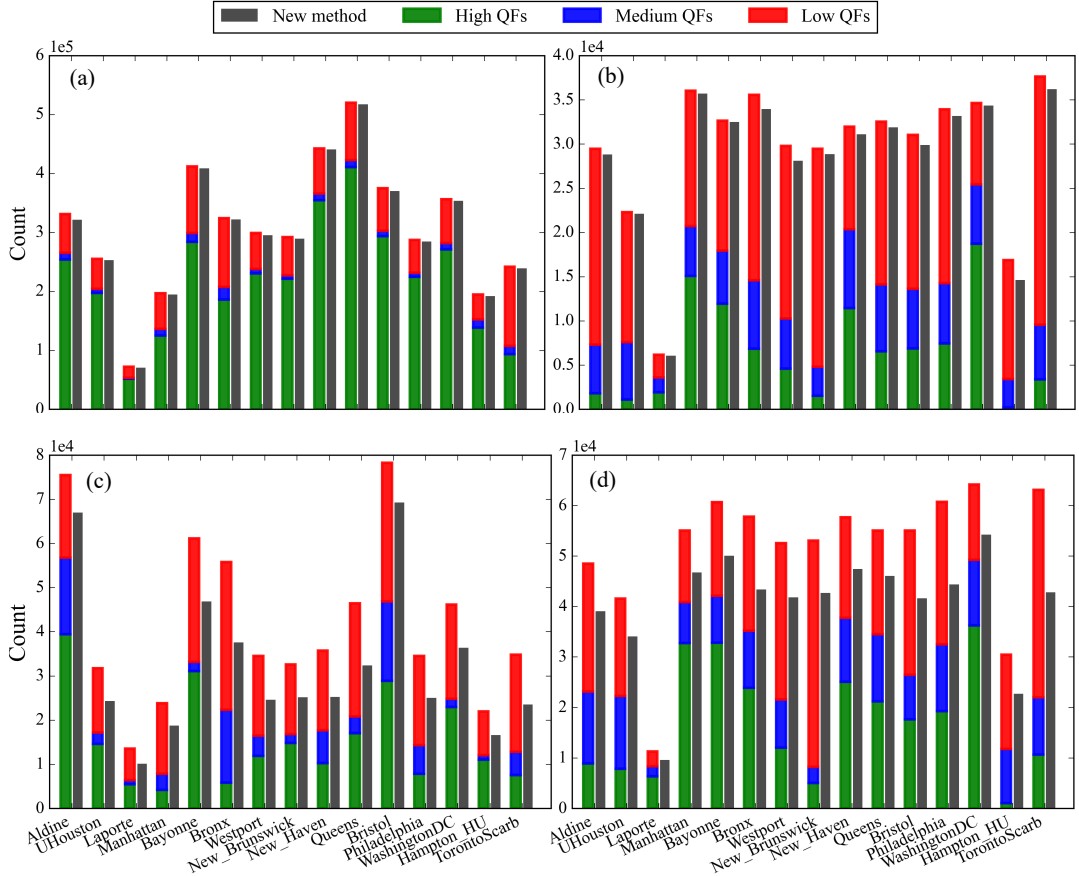

**Figure 11.** Quality flags (QFs) for each Pandora site in 2021-2022 (left bars) colored by green
(high), blue (medium), and red (low) for (a) NO$_2$ DS, (b) NO$_2$ SS, (c) HCHO DS, and (d) HCHO SS. Adjacent grey bars show total observation counts after applying the new filtering method.





### 3.1 Independent assessment of Pandora scientific data quality with HCHO:$O_3$ analysis

Recent work has shown a strong correlation between surface $O_3$ and column HCHO (Schroeder et al., 2016; Travis et al., 2022). This relationship is used here to test the effectiveness of the new filtering method on Pandora HCHO column observations for the University of Houston Pandora site during the TRACER-AQ period (September 2021). High-resolution (5-min) collocated surface ozone measurements were obtained from the Texas Commission on Environmental Quality. The

Pandora HCHO column measurements in SS and DS modes are temporally matched (within 5 minutes) with surface ozone data between 10:00 and 18:00 local time when photochemistry is most active. This analysis compares results using only high-quality flagged Pandora data with results using Pandora data filtered with the new method.

Figure 12 shows the relationship between the HCHO column and surface ozone using DS and SS observations. When the new filtering method is applied, the $R^2$ substantially increased in SS mode from 0.42 (Figure 12a) to 0.57 (Figure 12c) with a five-fold increase in the number of data points (181 to 955). In DS mode, $R^2$ remained the same (Figure 12b and 12d) but with over a two-fold increase in data (275 to 696). Similar improvements in correlation and data quantity are obtained

for other months outside the TRACER-AQ period that are not shown here.

    This analysis is expanded in Table 1 to include Pandora sites that are located within 500 m of an EPA regulatory monitor reporting hourly ozone. Data is for the month of September 2022 which had the best data availability. Table 1 summarizes the resulting correlation between hourly

averaged column HCHO in SS and DS modes with hourly surface ozone. Generally, similar or improved correlation and increased data availability (14 – 163%) were observed after applying the new filtering method. However, over the New Brunswick site in DS mode, a lower correlation (0.60 to 0.52) was observed after applying the new method. This was caused by the new method capturing an ozone event (>60 ppb) that appeared to be decoupled from high HCHO columns on

September 19, 2022. This independent analysis shows that recovered data can both improve the analysis of the HCHO:$O_3$ relationship and provide confidence in the scientific value of the new filtered dataset.






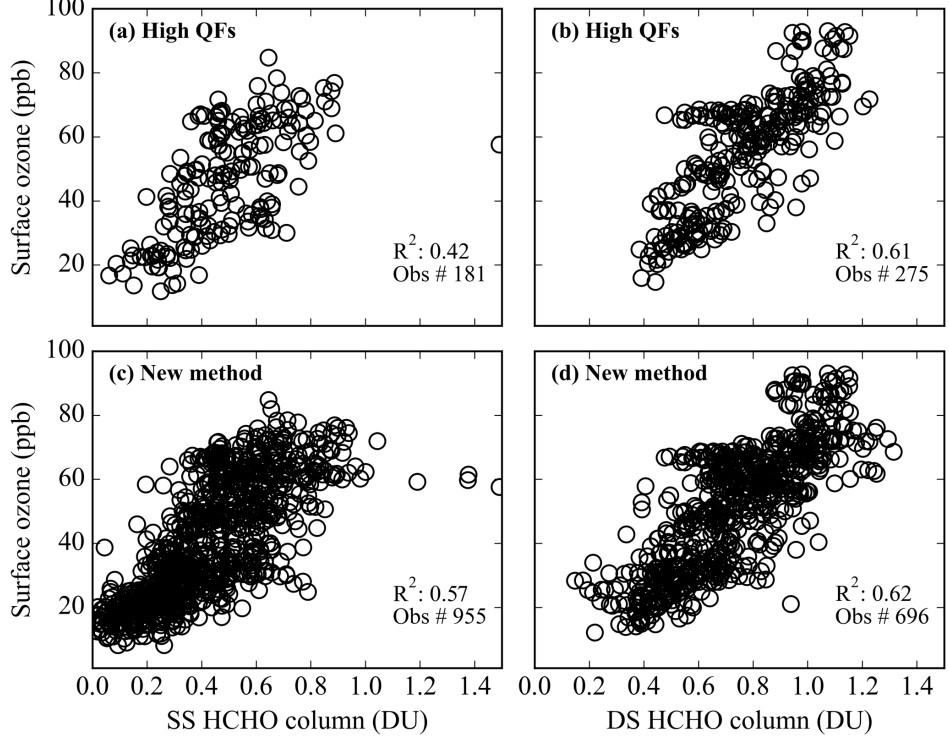

**Figure 12.** Relationship between Pandora HCHO column and surface ozone observations at the University of Houston site during TRACER-AQ in September 2021. Scatter plot between sky-scan HCHO column and surface ozone for (a) only high-quality flags and (c) data with new
filtering method. Scatter plot between direct-sun HCHO column and surface ozone for (b) only high-quality flags and (d) data with new filtering method. HCHO column is given in Dobson units (1 DU= $2.69 \times 10^{16}$ molecules cm$^{-2}$).







**Table 1.** Correlation ($R^2$) between hourly surface ozone and HCHO observations from direct sun and Sky scan for high-quality data flags alone and for data using the new filtering method. Analysis is for four Pandora sites during September 2022. Each site is located near (within 500 m) an ozone monitoring instrument. The number (n) is the available hourly matchups for each site.

| Site | Direct-sun | | Sky-scan | |
|---|---|---|---|---|
| | High Quality Flags | New Filtering Method | High Quality Flags | New Filtering Method |
| Manhattan | $R^2$=0.36, n=59 | $R^2$=0.43, n=156 | $R^2$=0.62, n=152 | $R^2$=0.64, n=164 |
| Westport | $R^2$=0.47, n=99 | $R^2$=0.55, n=165 | $R^2$=0.56, n=115 | $R^2$=0.57, n=189 |
| New Brunswick | $R^2$=0.60, n=154 | $R^2$=0.52, n=211 | $R^2$=0.61, n=222 | $R^2$=0.63, n=246 |
| Bristol | $R^2$=0.71, n=109 | $R^2$=0.67, n=209 | $R^2$=0.74, n=183 | $R^2$=0.72, n=241 |

**3.2 Independent assessment of Pandora scientific data quality with airborne remote sensing**

GEO-CAPE airborne simulator (GCAS) observations (Nowlan et al., 2018; Judd et al., 2020) provide an independent dataset of column $NO_2$ and HCHO observations to compare against Pandora over Houston, TX during the TRACER-AQ period (September 2021). Looking downward as it flies overhead at ~8.5 km, GCAS measures a partial column from the aircraft altitude down to the surface that includes the polluted boundary layer and a large portion of the less polluted free troposphere. Figure 13 shows the comparison of DS and SS observations with GCAS $NO_2$ and HCHO. To minimize the temporal and spatial mismatch between the Pandora and GCAS observations, GCAS was spatially filtered to include the nearest cloud-free GCAS pixels within 2.5 km of the Pandora location, and Pandora was temporally filtered to include observations in a 15-minute window centered on the time of the GCAS overpass.

Both the DS and SS $NO_2$ columns are well-correlated with GCAS ($R^2$ = 0.78 and 0.64, respectively). Some scatter can be attributed to GCAS uncertainties being on the order of 25% (Judd et al., 2020; Nawaz et al., 2024) from a priori uncertainties related to surface reflectivity and trace gas profile. For DS $NO_2$ columns, there is a clear offset with Pandora columns exceeding GCAS values. The slope near unity and intercept of $3\times10^{15}$ molecules cm$^{-2}$ is consistent with the stratospheric amount that would be absent in the GCAS observations below the aircraft. Despite a





few outliers (Figure 11b), there is no notable systematic bias between GCAS and SS $NO_2$. This is consistent with the SS $NO_2$ observations being limited to the most polluted lower few kilometers of the atmosphere. It could be argued that GCAS values tend to be slightly higher than SS observations of $NO_2$ due to the larger altitude range observed.

The correlations for GCAS and Pandora HCHO columns also show general agreement ($R^2 = 0.50$ and 0.51) but are weaker primarily due to the greater uncertainty in HCHO measurements for both instruments. For HCHO, the bias is in the SS rather than DS observations. This is due to the lack of significant HCHO in the free troposphere and stratosphere; thus, total columns associated with DS observations are more consistent with GCAS. Since HCHO is photochemically created, it tends

to have a weaker vertical gradient in the polluted boundary layer compared to $NO_2$ which is much more weighted towards the surface where it is emitted. Thus, the limited altitude range of SS observations are more likely to miss some fraction of lower atmospheric HCHO and be biased low with respect to both DS and GCAS HCHO.

Overall, these comparisons are encouraging for both GCAS and Pandora with differences that are

consistent with expectations based on the differences in DS and SS viewing sensitivities.




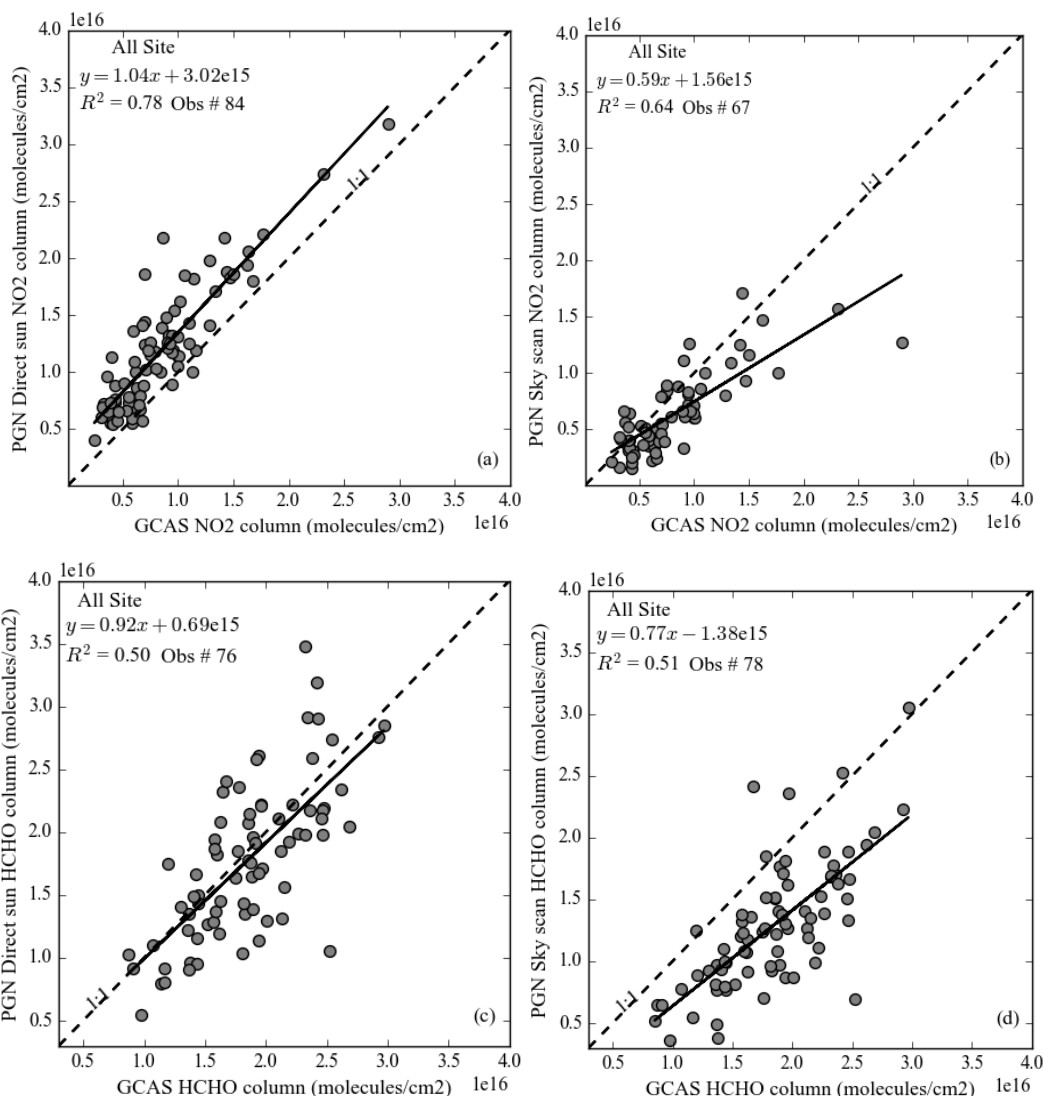

**Figure 13.** Comparison of airborne GCAS trace gas columns with and Pandora columns of $NO_2$ (panels a and b) and HCHO (panels c and d) during TRACER-AQ for Houston Pandora sites (Aldine, La Porte, and University of Houston). Comparisons are shown for both direct-sun (panels a and c) and sky-scan (panels b and d) modes.



### 3.3 Direct-sun and sky-scan intercomparison of column amounts

Given that the Pandora instrument spends significant time in both DS and SS modes, maximum data availability for scientific applications would be achieved by further addressing the causes of differences in their resulting measurements. DS and SS column measurements are expected to

differ in their absolute amount due to limitations in the vertical sensitivity (for SS measurements), retrieval method (Herman et al., 2009; Frieß et al., 2019), and physical differences between tropospheric and stratospheric amounts. SS columns generally profile from the surface up to 2-4 km while DS columns observe the total column through the entire atmosphere. Due to its short lifetime and major secondary source from chemistry, the HCHO column has a weak gradient from

the surface through the depth of the boundary layer (Schroeder et al., 2016; Crawford et al., 2021). This suggests that the DS HCHO total column should be greater than the SS lower tropospheric column, and the difference should be largely dependent on the vertical sensitivity of the SS measurement, which most often ranges from the lowest 2-4 km of the atmosphere. For the $NO_2$ column, the stratospheric contribution is the largest influence on the differences between SS and

DS column retrievals. To obtain a tropospheric DS $NO_2$ column, the $NO_2$ stratospheric column can be subtracted from the DS $NO_2$ total column using the stratospheric climatology provided by the PGN in their product based on Brohede et al., (2007).

Figure 14a shows the mean differences between the DS and SS column for HCHO and $NO_2$ over the 15 Pandora sites for 2021-2022. The bias for HCHO on average is $3.4 \times 10^{15}$ molecules cm$^{-2}$

with a larger bias at Hampton University in Virginia and New Haven, Connecticut ($>5 \times 10^{15}$ molecules cm$^{-2}$) and the smallest bias at Bayonne, New Jersey. The mean HCHO column in DS mode over these 15 sites averaged about $1 \times 10^{16}$ molecules cm$^{-2}$ making the bias between DS and SS mode approximately 35% of the column.

For $NO_2$, the mean difference was calculated after removing the stratospheric contribution (see red

line in Figure 14a, Figure 14 is without removing stratospheric contribution). The remaining bias was less than $1 \times 10^{15}$ or about 20% of the tropospheric column ($5 \times 10^{15}$) on average. Higher average $NO_2$ columns ($>10 \times 10^{15}$ molecules cm$^{-2}$) were observed in New York (Manhattan, Bronx, and Queens) which also had three of the four highest biases observed. The highest bias was at Bayonne and appears to be anomalous, demonstrating that bias comparisons across the network are a useful

tool for identifying potential problems at specific sites. In this case, Bayonne's large bias in $NO_2$



between SS and DS deserves a closer look by the site operator and may be related to a reference spectrum or calibration issue.

Figure 14 also shows how the bias between DS and SS for $NO_2$ (b) and HCHO (c) changes with solar zenith angle (SZA). One reason to expect changes in the bias with SZA is the larger

uncertainty in AMF calculations and larger stray light contribution at high SZA (Herman et al., 2009). However, no consistent patterns in bias behavior versus SZA was found for either HCHO or $NO_2$. Pandoras at Aldine and the University of Houston showed increasing biases with increasing SZA for HCHO but the bias decreases with SZA at New Haven and Toronto and showed no variations with SZA for the New York Pandora instruments. The $NO_2$ column biases as a

function of SZA showed no notable variation and were mostly constant around a mean value. Additionally, stray light is minimized in Pandora DS retrievals by subtracting the average signal below 290 nm from the spectra and using other correction methods (Cede et al., 2021). However, residual stray light could contribute to large difference between DS and SS HCHO columns. Plotting the residual stray light as a function of SZA (Figure S1) showed that it decreased or

remained constant (~ 0.3%) with increasing SZA, which suggests stray light contribution to the column might not increase significantly at higher SZA. There is also no evidence for a seasonal dependence in the difference between SS and DS observations (Figure S2 for the University of Houston), unlike that reported for the total ozone biases between Pandora and Brewer observations (Tzortziou et al., 2012).

Finally, the importance of differences in pointing azimuth and solar azimuth angle was investigated since 1) the Pandora SS analytical method may be less accurate when the solar azimuth and pointing azimuth are close (Cede et al., 2021), and 2) when the difference is large, the retrievals may be sampling different air masses. Figure S3 shows no systematic biases as a function of the difference between the pointing azimuth and solar azimuth across all 15 sites. This

suggests that retrieval uncertainties and vertical sensitivity are most likely responsible for the mean biases between the SS and DS columns. This allows for the possibility of combining the two datasets for higher temporal coverage.

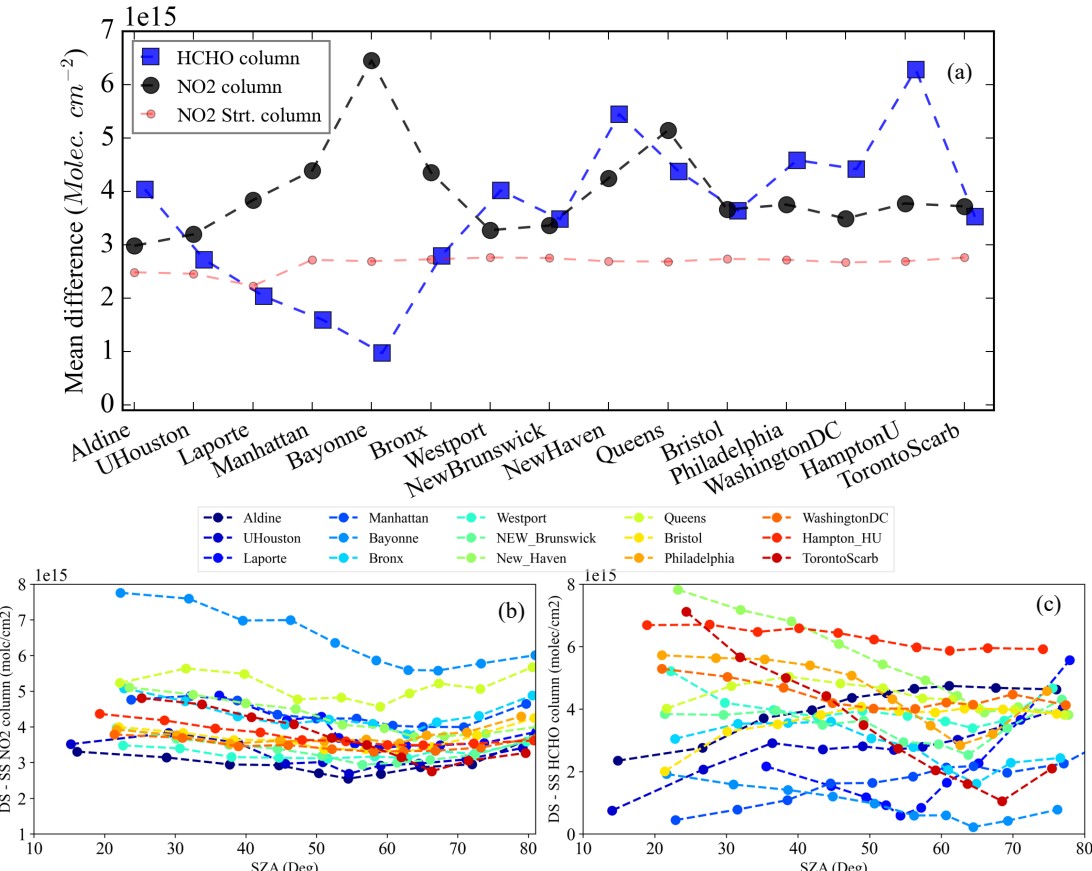

**Figure 14.** (a) Mean differences between the DS and SS column for HCHO and NO$_2$ over the 15 Pandora sites for 2021-2022. The NO$_2$ stratospheric column for each site is also provided. Bottom panels show the variation of mean column difference with SZA for NO$_2$ column (b) and HCHO columns (c), respectively.

### 3.4 Temporally combining DS and SS data

Section 2.2 discussed the standard schedule for Pandora observations with alternating measurements in the DS and SS modes and use of UV and open filters for the HCHO and NO$_2$ observations, respectively. The DS NO$_2$ retrieval is relatively fast compared to DS HCHO measurements which are less frequent. For SS measurements, both HCHO and NO$_2$ are comparable and take much longer than DS measurements (Figure 2c). In order to be combined, these time differences as well as biases diagnosed in the previous section must be addressed.





With the goal of creating a more continuous data record of hourly HCHO columns from Pandora suitable for comparison with hourly in situ measurements of surface ozone at EPA sites, Equation 1 combines data for DS and SS modes by correcting for the mean bias and then accounting for the difference in measurement times. Each column is weighted by the ratio between measurement time in each mode and total measurement time during the hour for both DS and SS modes. DS values for $NO_2$ column must first include subtraction of the stratospheric $NO_2$ contribution. SS observations are corrected by adding the mean bias value (Figure 14a). The summation of weighted values results in an hourly Pandora HCHO column.

$$X_{combined} = \sum_{i=1}^{1hr} X_{DS}[i] \times \frac{t_{eff}[i]}{Tot} + \sum_{j=1}^{1hr} \{X_{SS}[j] + MB\} \times \frac{\Delta t[j]}{Tot} \qquad \text{Eq. 1}$$

where $t_{eff}$ is the effective measurement time for each DS observation (see column 3: "Effective duration of measurement" in the Pandora data files), $\Delta t$ is the estimated measurement time for each SS observation (see Section S1.2 for details), Tot is the total integrated measurement time in a 1-hour period for both DS and SS measurements, $X_{DS}[i]$ is direct-sun column and $X_{SS}[j]$ is sky-scan column for the $i^{th}$ and $j^{th}$ measurements of each hour, respectively. MB is the mean bias of HCHO and $NO_2$ column between DS and SS measurements.

Table 2 shows the performance of the relationship between HCHO and hourly surface ozone at the University of Houston during September 2021 for SS, DS, and the combined dataset. The hourly correlation remains similar compared to high resolution data (5 min) as discussed in Figure 12, however the number of observations is lesser. The combined data set with new filtering method increased the temporal coverage of Pandora observations on the hourly data compared to using only one mode and high-quality flagged data. Applying the new filtering method to the combined data results in the greatest data coverage with better or equivalent correlation than for other subsets of data.



**Table 2.** Correlation ($R^2$) between hourly surface ozone and column HCHO from Sky-scan, Direct-sun, and combined data at the University of Houston site during TRACER-AQ (Sept 2021). Results include data with high quality only and with the new filtering method. The number of analyzed hours (n) is given for each case.

| HCHO:$O_3$ | High QFs | New filtering method |
|---|---|---|
| Sky-scan | $R^2$=0.41, n=132 | $R^2$=0.61, n=234 |
| Direct-sun | $R^2$=0.59, n=102 | $R^2$=0.58, n=191 |
| Combined data | $R^2$=0.50, n=162 | $R^2$=0.62, n=240 |

## 4 Summary and conclusions

The growing Pandonia Global Network of Pandora spectrometers provides a ground-based network of remote sensing observations to both validate satellite trace gas retrievals and understand local air quality. Pandora provides information on the diurnal variability of multiple species (e.g., $NO_2$ and HCHO) using two independent viewing geometries: direct-sun (DS) and sky-scan (SS) mode. The Pandora quality assurance procedure recommends using high-quality flagged data, exercising caution with medium-quality flagged data, and not using low-quality flagged data. Unfortunately, high quality data can often be only a small fraction of the total set of observations, except for DS $NO_2$ columns.

This work presents a new filtering method developed and tested as an alternative to using quality flags to identify scientifically useful observations. The method keys on the range of independent uncertainty values for Pandora data flagged as high quality. Given the large overlap in independent uncertainty for high-, medium-, and low-quality flagged data, the $\mu+3\sigma$ value of independent uncertainty for the high-quality flagged data defines a cut-off that preserves almost all high-quality data and includes a large amount of medium- and low-quality flagged data. For a much smaller number of data, wrms > 0.001 (normalized root mean square of the fitting residuals weighted with



the independent uncertainty) and SS MHzD > 20 km (maximum horizontal distance) should also be filtered out. A final step in the method is to restore observations with a percent uncertainty of less than 10%.

Given the independence and contemporaneous nature of DS and SS observation modes, correlation of adjacent (within 5 min) observations between modes provides evidence for scientifically useful data in the medium- and low-quality flagged data. Unfiltered observations already exhibit some degree of correlation for data with medium- and low-quality flags, but for filtered data, quality flags appear to have no influence on correlations for various pairing of DS and SS quality flags. This is particularly important in that the method can recover as much as 90% of data that would have previously been discarded.

The strong relationship between column HCHO and surface ozone provided a method for testing the new filtering method. At the University of Houston and four additional sites with nearby ozone monitors, filtered data resulted in a similar or improved relationship between column HCHO and surface ozone with 14-63% more data than using high quality data alone.

Airborne GCAS observations from the TRACER-AQ campaign in Houston provided an independent set of $NO_2$ and HCHO column observations to compare against filtered Pandora DS and SS observations. The two datasets correlated well, but biases were also observed related to differences in the total column DS observations versus SS observations that have sensitivity limited to the lowest 2-4 km of the atmosphere.

Combining SS and DS data provided better hourly data coverage after correcting for biases between the two modes. Across 15 Pandora sites in the domain of recent NASA field campaigns, the HCHO column biases ranged from $0.8\times10^{15}$ molecules $cm^{-2}$ to $6\times10^{15}$ molecules $cm^{-2}$ (8 to 60% of the average column of $1\times10^{16}$ molecules $cm^{-2}$). The $NO_2$ tropospheric column biases were less than $1\times10^{15}$ molecules $cm^{-2}$ (20% of the average column of $0.5\times10^{16}$ molecules $cm^{-2}$ except for at the Bayonne, NJ Pandora). The biases in $NO_2$ and HCHO between DS and SS observations were examined for trends with solar zenith angle (SZA), viewing azimuth angle, and residual stray light. No systematic behavior in the biases were found.



A more continuous hourly Pandora dataset was constructed for comparison with hourly surface ozone. This dataset required correcting the site-specific biases in DS and SS observations as well as weighting the observations based on their different measurement durations. The combined hourly dataset showed similar or improved correlations with surface ozone with a 30-40% increase in hourly data coverage.

The proposed filtering method and improved representation of hourly average conditions by combining DS and SS data offer the community useful strategies for using Pandora observations to their fullest potential for validating geostationary satellite observations and informing local air quality.

*Data availability*. The Pandora data is available at PGN website (https://www.pandonia-global-network.org). The airborne GCAS observation during TRACER-AQ period can be obtained from https://www-air.larc.nasa.gov/cgi-bin/ArcView/traceraq.2021. Surface ozone data at hourly and 5-minutes resolution is obtained from the EPA (https://www.epa.gov/ground-level-ozone-pollution) and ARM-TRACER-AQ (https://adc.arm.gov/discovery/#/results/s::tracer%20tceq) website, respectively.

*Author contributions*. PR, JHC, and KRT designed the study and PR performed the data analysis. PR and KRT interpreted the results and wrote the initial draft of the paper. LJ, JHC, KRT, JS, LV, MAGD, AW, EB and TFH provided significant conceptual input to the design of the manuscript and the improvement of the manuscript. All authors discussed the results and analysis.

*Competing interests.* At least one of the (co-)authors is a member of the editorial board of Atmospheric Measurement Techniques.

*Acknowledgements*. We thank the PI(s), support staff and funding for establishing and maintaining the 15 Pandora sites of the PGN used in this investigation, specifically we thank Jimmy Flynn, Maria Tzortziou, Nader Abuhassan, John Anderson, and Vitali Fioletov. We thank the ESA and NASA joint PGN for Pandora data processing and data disseminations. We thank NASA GFSC for Pandora calibration, in particular Apoorva Pandey and Bryan Place. NASA Postdoctoral Program at the NASA Langley Research Center, administered by ORAU under contract with NASA acknowledged for funding PR. LMJ and KRT acknowledge NASA grant

80NSSC24M0094. We acknowledge the GCAS groups for the TRACER-AQ 2021 filed campaign datasets. We thank EPA and TCEQ for surface ozone observations.

***Disclaimer***: The research described in this article has been reviewed by the U.S. Environmental Protection Agency (EPA) and approved for publication. Approval does not signify that the
contents necessarily reflect the views and the policies of the agency nor does mention of trade names or commercial products constitute endorsement or recommendation for use.

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
