# Peer review of "Maximizing the Scientific Application of Pandora Column Observations of HCHO and NO2"

_Atmospheric Measurement Techniques, 2024_

## Author Comment (AC1)

**Response to comments on "Maximizing the Scientific Application of Pandora Column Observations of HCHO and NO₂."**

We thank the reviewers for comments and suggestions that have helped to improve and clarify our paper. The manuscript is suitably revised by incorporating their suggestions and comments. We are also thankful to the editors for their time. Comments from reviewers are in black, responses are in blue, and new text added to the manuscript is in bold blue. The page and line numbers referenced correspond to the track changes document.

**RC1**

**General Comments**

**Overview**

The presented manuscript proposes an alternative data flagging procedure to the standard PGN flagging for HCHO and NO2 column densities, retrieved from MAX-DOAS and direct sun measurements, in order to increase the data amount that can be used for scientific studies. As such, the topic of the manuscript is highly important for users of PGN data-products. This can help the data-users and readers of this manuscript to better understand the standard flagging, and most importantly to apply their own filter criteria with the presented approach, or even go beyond. The authors use the linear correlation coefficient as a metric to validate their novel approach for both species, although the focus and interest is more on HCHO. The correlation of HCHO to surface O3, and airborne data for both HCHO and NO2 are presented as case studies.

The motivation to increase the sample for scientific analysis is certainly important, but the reason why data are flagged still needs to be taken into account. Unfortunately, the main part and the supplemental review of the quality flags is described rather vague, with both missing parts and wrong statements of the current flagging. Therefore, the manuscript would highly benefit from a more in-depth analysis of the standard quality flags to highlight the driving quality indicators which lead to the flagging, and the additional corrections about the current flagging.

We thank Manuel Gebetsberger for being a referee and providing very useful comments and constructive suggestions. We have now added a discussion on the triggers for data flagging from L1 to L2fit to L2 in supplementary section S1.3.

We have added the following paragraphs for data flagging in the revised manuscript in page 8 lines 196-206.

**These quality flags are assigned in various stages from L1 (raw data) to L2fit (spectral fit data) to L2 (processed data) for ensuring data quality through multiple checks at each stage. At the L1 stage, data is flagged into either low- or medium-quality based on instrument-related issues such as excessive dark counts, detector saturation, dark count differs significantly from the dark map for too many pixels, different effective temperature, and unsuccessful dark background fitting. In the L2fit stage, where spectral fitting is performed, data is further flagged based on factors including the quality of the fit, the wrms limit (normalized rms of fitting residuals weighted with independent uncertainty), and wavelength shift. Finally, at the final retrieval L2 stage, factors including retrieval error and atmospheric variability are used to flag data into the medium or low-quality.**

**Title**

The title is misleading in terms applicability to which PGN data product. The presented study focused on HCHO and NO2, but with a strong focus on HCHO. However, the Pandora data pool also covers O3, SO2 and H2O, which have not been demonstrated in the manuscript. For both O3 and SO2 there are no MAX-DOAS data products available, which limit the presented combined-approach to HCHO and NO2. H2O would be available by both the direct sun and MAX-DOAS measurements, but was not considered here. Therefore, the presented approach is not generic enough to be applied for all Pandora data products, which should be properly reflected in the title.

Thank you for raising this concern. To make the title clearer we have revised the title to **"Maximizing the Scientific Application of Pandora Column Observations of HCHO and NO$_2$".**

**Comment On Quality flags**

The quality flags are propagated from L1, to L2Fit, to L2 and end up in the different clusters for high, medium, low data quality, that can be un-assured, assured, or unusable. This means if a single retrieval is identified as low data quality on the L1 side, it cannot be of better quality on higher levels. If the number of dark cycles is already too low, or saturated data occurred already on the L0 side, or the spectrometer temperature is too far away from the characterized temperature in the laboratory, data will be flagged into medium or low quality already. The same applies on higher levels. If for instance the L1 retrieval is of high quality, but the spectral fitting wrms is exceeded due to a spectral signal which cannot be captured by the retrieval polynomials, data can be flagged into the categories based on the threshold which is exceeded.

The thresholds for some of the quality indicators come from the Gaussian Mixture Regression model approach. This approach is applicable to individual datasets, such as P25s1 HoustonTX. However, the PGN flagging does not use instrument-specific flags and uses a PGN average over multiple datasets. This could indeed lead to some datasets flagged to strict and some to weak. Has this approach been tested, to use for instance a HoustonTX-specific threshold of the wrms to not by-parse L1 filter?

More general, the wrms is the quality indicator of the spectral fitting. By by-parsing this quality indicator as a flagging criteria, potential slant column biases introduced by spectral signals are ignored. The same applies for the unusable category of 20,21,22, which should not be ignored.

On the other hand, there might be quality indicators which are too strict and can lead to a filtering of valid retrievals, which is the motivation of this manuscript. It would be needed to identify those quality indicators. Maybe, there is one or two quality indicators which are responsible for the majority of the filtering of 'valid' retrievals. It would be interesting to see if the simple removal of them can already lead to the same effect as the proposed approach.

We have now added supplementary section S1.3 to discuss the flagging of each data point at different stages of the Pandora filtering criteria and corresponding triggers for each flag in the medium (1, 11) and low quality (2, 12) with details of recovered points using the new method.

Please find the summery for section S1.3 in manuscript at page 18-19 lines 360-374.

We also added the following sentences to the abstract on page 1, line 33-35 and to the conclusions on page 31, line 633-635.

**This method suggests that standard PGN criteria for atmospheric variability and normalized root mean squared error are too stringent as they are responsible for downgrading most of the recovered data.**

**Further analysis suggests that the standard PGN data flags for atmospheric variability and wrms are too stringent based on the quality of data recovered that exceed PGN criteria for those values.**

**Specific Comments**

Here I refer to the lines of the manuscript:

18 change "PGN standard quality assurance" to "PGN standard quality flagging", since the assurance part does not change the high, medium, or low quality categorization

Changed.

25 Have other uncertainty components been analyzed?

No, in our analysis of contemporaneous measurements of DS and SS only independent uncertainty will affect the correlation. The structured and common uncertainties apply over longer timescales that do not affect this analysis of short-term correlations.

26 Confusing statement of "independent uncertainty filter". Does this refer to the reported "independent uncertainty" component in the L2 file or the presented approach which uses the "independent uncertainty"?

Yes, it is the independent uncertainty reported in the L2 files. We have now revised it as '**After applying a filter on independent uncertainty**'.

56 "interferences" would refer more to an optical problem. With respect the Delrin-problem, it was HCHO that was measured and retrieved, but it was not the just atmospheric HCHO in the lightpath.

We have changed it accordingly in the revised manuscript page 2, line 61-62.

**outgas HCHO and interfere in its detection**

60 replace https://blickm.pandonia-global-network.org to https://www.pandonia-global-network.org/ since not all data-users do have a blickm account. And blickm is a monitoring tool without providing reports, software, or data to download.

Replaced.

65 LuftBlick with capital "B"

Changed.

75 With respect the direct sun HCHO retrieval I would additionally cite the ReadME:https://www.pandonia-global-network.org/wp-content/uploads/2023/11/PGN_DataProducts_Readme_v1-8-8.pdf since Spinei et al. is the originator of the MAX-DOAS retrieval but not of the direct sun.

Added.

91 At the end I would mention that the approach has been demonstrated solely for HCHO and NO2.

We have added the following line in the revised manuscript at page 3, line 94.

In this work, a data filtering method is presented for assessing suitability **of HCHO and NO$_2$ column data** beyond the standard quality flags provided by PGN.

92 H2O is also an official data product provided by the PGN, with rcodes wvt1 and nvh3 for direct sun and MAX-DOAS retrievals, respectively.

In the present work only HCHO and NO$_2$ column are explored.

105 Not all Pandoras are stabilized at 20, some are also measuring at 15, which highly depends on the location and environment where the instrument is set-up.

We have revised the line at page 4, line 111-112.

To minimize the dark current noise, the spectrometer is maintained at a stable temperature **of 15-20°C (based on the site location) using** a thermal electrical cooler.

111 latency correction is not applied in a characterization step, and also not characterized in the laboratory, since it would require to open the spectrometer and flip the CCD.

Removed.

112 stray light characterizations are not applied in the processor 1.8, which limits the stray light correction to the simple straylight method, which is to subtract the signal below 290 nm. The straylight correction matrix method currently not applied.

We have changed it in revised manuscript at page 4, line 118.

**simple** stray light **removal.**

161 Here the user might benefit from the information that the highest angle is used a reference in the spectral fitting. Which further means, that if this angle is contaminated by an obstruction (e.g. tree), a spectral signal could enhance the wrms and further the data product. What was the azimuth angle of all the datasets used in this study? Where the instruments looking in the same direction?

We have added the description in page 6, lines166-171.

In the SS mode, zenith measurement scans are routinely collected at specified azimuth angles, with the north or 0° being the preferred direction **for northern hemispherical sites.** The zenith angles scans can vary by site with the lowest scan typically occurring between 89° and 85° depending on viewing conditions available at the site to the highest zenith angle (0°), **which is utilized as the reference during the spectral fitting (details of different zenith angles are provided in section S1.2).**

185-189 uncertainties are not used in any part of the processor 1.8 flagging procedure. It is true that the "total uncertainty" of the processor 1.7 was used which is the independent uncertainty. But it was removed from the flagging criteria in 1.8. The reason was that this parameter was too dependent on the instrument's sensitivity and schedule/routines the Pandora is measuring, since longer exposure times typically have larger independent uncertainties. However, processor version 1.8 provides a detailed uncertainty budget which might be of interest also as a decision criteria which data to use.

We have now removed uncertainties from the list of quality flag indicators and added the more correct listing of quality flag indicators in page 8 lines 196-208, and also replied above at "Comment On Quality flags".

190-202 What is the reason for "much of the data unavailable"? Which parameter is the driving quality indicator?

We have exhaustively analyzed the quality flags in supplementary section S1.3 enabling us to now be more specific about which quality indicators are responsible. See response to "Comment On Quality flags".

226 It is expected that the independent uncertainty has an overlap in all quality flag categories, since it is not reflecting any issues on L1 site (e.g. small number of cycles) or L2Fit side (spectral features which cannot be captured). It would also not reflect an air mass factor error on the L2 side, if the instrument has been using the wrong PC time for example. This problem is highly

impacting the L2 columns in terms of the diurnal shape which is of interest for satellites like TEMPO. Pandora25s1 at HoustonTX has this problem. Here the periods have been categorized as unusable (20,21,22) by the quality assurance part. How is the presented approach accounting for such situations if no quality assurance has been applied on the dataset? Because this would is not reflected in the independent uncertainty, because the instrument can still be properly aligned and looking into the sun.

While we cannot address such a situation prior to quality assurance, when including unusable data in our analysis, correlations are poor.  If such a condition existed, we would not expect our method to succeed. It is important to note that when applying this filtering method, it is still incumbent on the data user to examine the data to see if correlations improved. If they don't, other problems are impacting the data. We have changed the text on page 11, line 259 to reflect the need for scrutinizing the data after the filter is applied.

In this method, Pandora data having independent uncertainty less than the cut-off limit are **further scrutinized** for scientific utility regardless of their data flag.

255 Figure 6. How is the independent uncertainty related to the atmospheric variability parameter?

The independent uncertainty and atmospheric variability are seemingly unrelated. This is shown in supplementary section S1.3.

265 How is this threshold defined and what is the objective approach behind?

Our approach was empirical. We have revised this in the manuscript at page 13, line 289-293 to say

**we empirically determine that these outliers are associated** with **unusually high values for DS and SS data of high wrms and for SS data of** maximum horizontal distance (MHzD). **These outliers** can **be removed** with thresholds for wrms of 0.01 and MHzD of 20km.

270-272 Is this improvement related to the data removal due to the wrms < 0.01

No. the wrms step only address the few outliers. Most of the data improvement comes from the independent uncertainty filter.

Figure10 as soon as MAX-DOAS comes into the recipe, the approach is not applicable for O3, SO2. It would also be needed to analyze H2O to demonstrate the applicability in a broader context.

Our study is only focused on HCHO and $NO_2$ columns, which are both available in DS and SS mode and no work is performed for other trace gases. To make it clear now we have changed the title to "**Maximizing the Scientific Application of Pandora Column Observations of HCHO and $NO_2$**".

322-324 This strong increase is great! However, since the flagging approach is not taking into account L1 related problems, nor potential slant column biases in the spectral fitting (covered by the wrms), some justification is missing if each retrieval is really usable or not. Is this increase attributed due to by-parsing one or two of the standard flagging criteria already? And if, which are those?

This concern is now addressed by the addition of supplementary section S1.3. See response to "Comment On Quality flags".

355-360 Is the $R^2$ the proper measure to demonstrate the applicability? Under the assumption to have a linear correlation, the correlation and $R^2$ should remain similar if 100 or 1000 datapoints are used. If the $R^2$ differ significantly, could this imply to have an undersampling or wrong assumption of the relationship? Can the $R^2$ between two different populations be compared directly? The relationship in Figure 12 implies a little bit to be non-linear for the MAX-DOAS columns. Can you provide some uncertainty range of the $R^2$, maybe by cross-validations or bootstrapping approaches? Or is there any expected correlation from literature between HCHO and surface O3 which supports a certain $R^2$ value where the sample should converge?

$R^2$ is indeed the proper measure as it determines the explainable variance in ozone. The analysis shows clearly that the explainable variance stays the same or improves due to the ability to include more data in the analysis. The quantifiable increase in data quantity for the period of a field campaign is always advantageous, regardless of whether it leads to a better characterization of the $O_3$:HCHO relationship. Advancing research on this relationship is the entire point of our ongoing research.

487 What is meant by other methods?

Removed.

524 Is the means bias value showing some seasonality due to different mixing heights in summer and wintertime? This would mean in summer the MAX-DOAS would not see a larger fraction of the total column compared to wintertime. This could indicate a smaller mean bias in winter than in summer.

While we do understand why you might have expected this, we have already shown in Figure S2 that there is no seasonal trend in the bias.

Table 2 Can you provide any uncertainty ranges of the R^2?

No, there is no practical way to determine uncertainty ranges of the $R^2$ values, and we have never seen it done. In fact, $R^2$ is related directly to the uncertainty in the slope and intercept of the regression. Thus, an increase in $R^2$ should be considered a decrease in uncertainty.

565 is the wrms threshold of 0.001 site-specific or generally applicable? How is this 0.001 related the the wrms threshold of 0.01 reported in Figure10 and on line 265?

We find the wrms threshold works fine as a generally applicable value. The value of 0.001 on line 689 was a typo, which is corrected in revised manuscript. Thank you for catching this mistake.

575 It would be very interesting to see why so many datapoints are discarded. If this is related to 1 or two quality indicators. I encourage the authors to look into the L2 file, where all the needed information is reported (see example of L2 flag propagation).

We agree. These details are now available in supplementary section S1.3. Also see response to "Comment On Quality flags".

---

## Author Comment (AC2)

**Reviewer #2**

We thank Reviewer 2 for their comments and suggestions. The point-by-point responses are provided below, where comments from reviewers are in black, responses are in blue, and new text added in manuscript is in bold blue.

Rawat et all in their manuscript "Maximizing the Use of Pandora Data for Scientific Applications" present a methodology to increase the amount of "scientifically usable" columnar NO2 and HCHO data from Pandonia Global Network by applying different from PGN standard filtering criteria. The approach consists in using an independent uncertainty (detector photon noise propagated to slant columns) threshold to eliminate poor quality data. This threshold is derived from the independent uncertainty distribution for high-quality flagged data as $\mu + 3\sigma$. The data are further filtered by nrms (> 0.01) and maximum horizontal distance estimation for tropospheric columns (>20 km), and restoring measurements with < 10% relative error. The filtering results are verified by conducting linear regression analysis between different combinations of standard PGN quality flagged tropospheric column vs total column data of NO2 and HCHO. The main assumption is that the data are "scientifically useful" if correlation $R^2$ is consistent for various flag combination of tropospheric column (scattered sky) vs direct sun measurements after filtering.

The focus of the paper is to better understand the quality of trace gas column measurements and "recover" PGN data potentially incorrectly labelled as low quality. This is a relevant topic for a publication in AMT considering the importance of PGN for satellite validation and air quality research. However, in current version this paper does not add any new knowledge about the quality of the measurements, and "physical" reasons for accepting more measurements.

We respectfully disagree with the reviewer. Comparing Pandora measurements from DS and SS that are fully independent of each other as well as Pandora measurements to surface in situ ozone that are also fully independent is an ideal way to demonstrate the value that can be gained from filtering Pandora data with our method. The independence of these measurements is fundamental 'physical evidence' of our approach.

We now better address the physical causes of high data loss in the standard PGN flags as overly stringent atmospheric variability and wrms thresholds which is fully explained in the new supplemental section 1.3.

Major comments:

The main assumption that the data are "scientifically useful" if linear correlation $R^2$ is consistent for various flag combination of tropospheric column from scattered sky vs total columns from direct sun measurements is not totally proven. While I agree that they have separate analysis "paths" they do not have to be correlated (e.g. sampling different air masses due to difference in observation geometries) and they can be correlated for wrong reasons (e.g effect of clouds and aerosols, observation geometry). Actually, the only times they could be correlated are under totally cloud free, homogeneous conditions and perfect instrument performance – the high-quality flagged data.

If two methods are measuring the same quantity in quick succession, the expectation of autocorrelation is fundamental. Therefore, these $R^2$ correlations are an indication that the measurements are valid.

The authors need to show how the parameter subset that goes into quality flag determination changes because of their filtering to convince that the resulting data are scientifically acceptable. There are certain metrics (e.g. wavelength shift) that have less impact on the DOAS fitting and air mass factor quality than others (e.g. clouds). The value of this paper would be to identify such main "drivers" of data quality based on a very detailed evaluation of instrumental and atmospheric uncertainties in PGN data.

This is now addressed in supplementary section S1.3, which was also requested by reviewer 1. The outcome that atmospheric variability (e.g., clouds) have no apparent bearing on data quality is a tremendous significance and raises the question of whether the atmospheric variability parameter is related to clouds at all.

In general, poor quality in direct sun DOAS fitting results can rise from instrumental problems (tracker pointing issues, coherent light interference, internal stray light, filter wheel issues,

spectrometer changes leading to wavelength and slit function drifts, etc, inaccurate location or time) and atmospheric (presence of the clouds, leading to change in photon path and spectral saturation, spatial stray light). Poor quality in scattered sky data is mainly due to presence of cumulus clouds at the higher scan angles, pointing at obstructions, presence of clouds in the reference spectrum and pointing close to the sun, changes in scattering conditions between the scan measurements etc. There are two parameters that reflect the data quality to the first order: nrms of the DOAS fitting residuals and relative column error. Nrms is instrument and fitting window dependent and thresholds can be determined from the fitting data. Also, nrms of 0.01 is a very large value for typical trace gas DOAS fits to be valid.

Our maximum limit of wrms < 0.01 was set empirically and our correlation between SS and DS indicates that these points are reasonable.

A lot of examples were provided on the data from Houston, Texas metropolitan area, a near coastal region with relatively high presence of partly cloudy conditions. The presented results of the proposed filtering suggest that more than 90% of scattered sky data were not impacted by clouds. This might be overly optimistic.

Sky-scan observations are almost never triggered to low or medium quality by atmospheric variability in the standard PGN flags (Table S4).

Direct sun HCHO depends on spectrometer stray light properties and/or some other potential optical interferences. As a result, caution should be taken when interpreting the measurements. For example, collocated instruments often will not produce the same HCHO total columns.

We appreciate the need for caution in interpreting the HCHO DS measurements. However, if the measurements are flawed, there should be no expectation of a correlation with surface in situ ozone. This relationship demands that we reassess the quality of the DS HCHO data. Nevertheless, we have discussed the residual stray light as a function of SZA in Figure S1, which showed it decreased or remained constant (~ 0.3%) with increasing SZA, which suggests stray light contribution to the column might not increase significantly at higher SZA. Also, differences between DS and SS in section 3.3 are analyzed along difference SZA, along different azimuth viewing in Figure S3 and for different seasons in Figure S2 and no specific behavior is observed.

It appears that there are interpolation errors in figures 8 and 9: constant Y values for changing X values.

There is no interpolation used in the present analysis. However, in figures 8 and 9 we see changing X for constant Y, as we match all the DS observations with the nearby SS observation within 5min.

To make it clearer now we have added a line in revised manuscript at page 9, line 226.

This enables an independent assessment of data quality by **taking advantage of the expected autocorrelation** of contemporaneous (within 5 min) DS and SS observations for different quality flag combinations.

I find the idea of using correlation improvement between column (HCHO) and surface (O3) measurements is a weak argument for selecting data quality of the measurements. The goal should be to derive this (surface to column) dependence based on the best quality data and not force it through selection of the data. Multiple studies have shown that column to surface ratios depend on a number of meteorological, emission and photochemistry conditions, so using it as an argument in favor of the new filtering might not convince the broader scientific community.

We respectfully disagree. Improved correlations do not happen by accident. Our filtering process clearly shows that there is improvement in the correlation between two entirely independent observables. Please explain how poor quality data could result in such an outcome.

GCAS measurements depend on surface reflectance, aerosol and trace gas profiles and need their own validation. These measurements are typically considered less accurate than ground-based measurements. Again, using them as a verification tool seems not appropriate.

We again respectfully disagree. Not only do we show correspondence between GCAS and Pandora, we show that the discrepancies between the two are entirely explainable. Comparing quantities attempting to measure the same variable with an eye for why they should or should not match is fundamental science. Good correspondence is unlikely to be accidental for two independent measurements.

It is important to note that we are not the first group to compare GCAS observations with another observational perspective. We have added a line in the revised manuscript at page 23, lines 458-460.

**Additionally, the GCAS tropospheric column measurements have been shown to strongly correlate with in-situ aircraft observations for $NO_2$ ($R^2=0.89$) and HCHO ($R^2=0.54$), with columns differences in magnitude within 10% (Nowlan et al., 2018).**

Missing detailed description of the standard PGN data flags and parameters that go into their determination

We have added the following paragraphs for data flagging in revised manuscript in page 8, lines 196-206 and detail Pandora data flagging propagation in supplementary section S1.3:

**These quality flags are determined in various stages from L1 (raw data) to L2fit (spectral fit data) to L2 (processed data) for ensuring data quality through multiple checks at each stage. At the L1 stage, data is flagged into either low- or medium-quality based on instrument-related issues such as excessive dark counts, detector saturation, dark count differs significantly from the dark map for too many pixels, different effective temperature, and unsuccessful dark background fitting. In the L2fit stage, where spectral fitting is performed, data is further flagged based on factors including the quality of the fit, the wrms limit (normalized rms of fitting residuals weighted with independent uncertainty), and wavelength shift. Finally, at the final retrieval L2 stage, factors including retrieval error and atmospheric variability are used to categorize data into the medium or low-quality.**

Need to provide more details for Figure S1: wavelengths, how this residual stray light was determined, etc

We discussed the residual stray light determination in the manuscript at page 27, line 543. We are not sure what else is needed.

---

## Referee Report (RR1)

**Review_amt-2024-114_v1**

Manuel Gebetsberger, LuftBlick

2/18/25

**Table of contents**

**General Overview**

The presented manuscript proposes an alternative data flagging procedure to the standard PGN flagging for HCHO and NO2 column densities retrieved from MAX-DOAS and direct sun measurements. The aim is to increase the amount of usable data for scientific studies. As such, the topic of the manuscript is important for users of PGN data products. This approach can help data users and readers of the manuscript better understand the standard flagging method and, most importantly, apply their own filter criteria using the presented approach—or even go beyond it.

The authors use the linear correlation coefficient as a metric to validate their novel approach for both species, although the primary focus is on HCHO. The correlation of HCHO with surface O3, as well as airborne data for both HCHO and NO2, is presented as a case study.

The manuscript also provides a more in-depth analysis of flagging propagation and identifies the most influential quality indicators responsible for data flagging. One parameter is indeed questionable in terms of whether it should even be used as a flagging criterion. The second parameter is based on spectral fitting RMS. While the authors use an empirical value, the PGN threshold appears to be too strict. However, users should be cautious about disregarding this entirely.

**Minor comments**

Line 160 The 0° direction is not necessarily the 'preferred' direction; it is simply a software default in the config file. Therefore, any alignment with an interesting air mass could be coincidental. Or, it may indicate that the instrument owner has not given much thought to the optimal measurement direction. The key message is that, regardless of where the instrument is scanning, the lowest elevation angle should not be obstructed by an obstacle.

Line 387 I still see the increase in $R^2$ as being more related to the actual non-linear behavior, which becomes more apparent as the dataset size increases. This can reduce noise in the data and stabilize $R^2$, which might be overinterpreted compared to the smaller sample (#181) in the SS data, where an outlier has a stronger influence. Since $R^2$ is a linear correlation metric, relying on it alone to assess correlation can be misleading. A larger $R^2$ may simply be an artifact of a larger dataset and does not necessarily imply a stronger linear relationship—especially in cases where the data already exhibit non-linearity and varying variance, as seen here.
To illustrate this, I have generated a small random dataset with both linear and non-linear relationships, along with some outliers. In the linear sample, the $R^2$ values remain similar, whereas in the non-linear sample with outliers, the $R^2$ suggests a 'substantial' improvement.

422 Building on the previous comment, in my opinion, the increased $R^2$ does not necessarily improve the analysis of the HCHO:O3 relationship itself. Rather, it appears to be an artifact of the sampling process, combined with the influence of outliers. However, the new filtering increases the dataset size significantly, making other underlying processes more apparent, which may allow for additional conclusions to be drawn.

```
Warning: package 'ggplot2' was built under R version 4.1.3

`geom_smooth()` using formula = 'y ~ x'`geom_smooth()` using formula = 'y ~
x'`geom_smooth()` using formula = 'y ~ x'`geom_smooth()` using formula = 'y ~ x'
```

[Figure]

Figure S2 For HCHO, there appears to be a slight positive slope, with the difference being around 0 at SZA = 10° (JJA) and approximately 0.4 at SZA = 78° (SON). Such an SZA dependency is not observed for NO2. Is this effect negligible and statistically insignificant for HCHO, or what could be the possible cause of this small slope?

**Comments on References**

Line 677: The provided link is not working. An updated version is available at: Blick Software Suite Manual v1.8.5.

Lines 684–691: The same report is referenced twice. I recommend using the latest version: PGN Data Products Readme v1.8.10 Please adjust the references accordingly in the manuscript.

Line 716: This reference appears to point to the same link mentioned in my previous comment, likely due to an incorrect copy in the bibliography file.

Line 720: Same issue as the previous two comments. The link appears to be broken and points to the wrong content.

Line 810: Broken link. Please update or remove as needed.